# Second-Generation Phage Lambda Platform Employing SARS-CoV-2 Fusion Proteins as a Vaccine Candidate

**DOI:** 10.3390/vaccines12111201

**Published:** 2024-10-22

**Authors:** Alexis Catala, Bennett J. Davenport, Thomas E. Morrison, Carlos E. Catalano

**Affiliations:** 1Department of Pharmaceutical Sciences, Skaggs School of Pharmacy and Pharmaceutical Sciences, University of Colorado Anschutz Medical Campus, Aurora, CO 80045, USA; alexis.catala@cuanschutz.edu; 2Department of Immunology and Microbiology, University of Colorado Anschutz Medical Campus, Aurora, CO 80045, USA; bennett.davenport@cuanschutz.edu (B.J.D.); thomas.morrison@cuanschutz.edu (T.E.M.)

**Keywords:** phage-like particles, viral nanoparticles, vaccine development, SARS-CoV-2, nanomedicine

## Abstract

The recent SARS-CoV-2 (COVID-19) pandemic exemplifies how newly emerging and reemerging viruses can quickly overwhelm and cripple global infrastructures. Coupled with synergistic factors such as increasing population densities, the constant and massive mobility of people across geographical areas and substantial changes to ecosystems worldwide, these pathogens pose serious health concerns on a global scale. Vaccines form an indispensable defense, serving to control and mitigate the impact of devastating outbreaks and pandemics. Towards these efforts, we developed a tunable vaccine platform that can be engineered to simultaneously display multiple viral antigens. Here, we describe a second-generation version wherein chimeric proteins derived from SARS-CoV-2 and bacteriophage lambda are engineered and used to decorate phage-like particles with defined surface densities and retention of antigenicity. This streamlines the engineering of particle decoration, thus improving the overall manufacturing potential of the system. In a prime-boost regimen, mice immunized with particles containing as little as 42 copies of the chimeric protein on their surface develop potent neutralizing antibody responses, and immunization protects mice against virulent SARS-CoV-2 challenge. The platform is highly versatile, making it a promising strategy to rapidly develop vaccines against a potentially broad range of infectious diseases.

## 1. Introduction

The coronavirus disease 2019 (COVID-19) pandemic was a global outbreak of a novel respiratory virus, severe acute respiratory syndrome coronavirus 2 (SARS-CoV-2) [1,2]. Cases in Wuhan, China were first reported in December 2019, and the rapid spread of the virus (>9000 cases, >200 deaths reported in 20 countries) prompted the World Health Organization (WHO) to declare a *Public Health Emergency of International Concern* by January 2020. Shortly thereafter, the outbreak was characterized as a pandemic with >750 k cases and >36 k deaths reported globally by March. The federal *Public Health Emergency for COVID-19* ended in May 2023, and the WHO estimated a total of >0.7 billion confirmed cases and >7 million reported deaths globally as of January 2024 (*COVID-19 Epidemiological Update, Ed. 163*). The swift engineering of RNA-based coronavirus (CoV) vaccines was instrumental in controlling disease severity; however, the requirement for multiple doses (a barrier for vaccine-resistant patients) and the need to maintain extreme and strict “cold-chain” conditions (a barrier to vaccine deployment) continue to pose significant limitations for current vaccines, especially in low-resource countries [3,4,5]. Thus, additional platforms are warranted to address issues related to rapid re-formulation, vaccine stability and the capacity to provide protection against both currently circulating viruses and those identified with a high pandemic potential.

The WHO COVID-19 Vaccine Tracker & Landscape database for March 2023 listed 183 vaccines in clinical development, the majority of which are based on protein subunit platforms, such as non-replicating viral vectors and virus-like particles (VLPs). Three of the seven VLP candidates in phase 3 clinical trials rely on particles derived from either bacteriophage (phage) AP205, a VLP vector or SARS-CoV-2 and are modified to display the receptor-binding domain (RBD) of the SARS-CoV-2 spike (S) glycoprotein S1 fragment or mutant S trimers [6,7,8]. We have developed a designer nanoparticle platform that employs phage-like particles (PLPs) derived from phage lambda that can be decorated *in vitro* with cocktails of wildtype and modified (genetically, chemically or mixed) phage lambda decoration proteins [9,10]. The particle surface can thus be functionalized with small molecules, synthetic polymers, nucleic acids and foreign proteins for diverse applications. Specific examples include the use of fluorescent proteins or chemical fluorophores for biomedical imaging and to study intracellular trafficking [11,12]; antibodies for targeted cell recognition and intracellular delivery of biologics [12,13]; and the multivalent display of viral antigens from respiratory viruses for use as a vaccine against both SARS-CoV-2 and Middle East Respiratory virus (MERS)-CoV infection [14].

Our ‘first-generation’ phage lambda vaccine platform depended on chemical cross-linking of mutant gpD protein to CoV RBDs for antigen display, which generated logistical concerns related to large-scale manufacturing. Namely, chemical cross-linking approaches can result in product heterogeneity and yields of the purified conjugate may be modest. Hence, to mitigate these issues and streamline the vaccine development workflow, we developed a second-generation platform using genetically engineered SARS-CoV-2 RBD and lambda gpD fusion constructs. PLPs decorated with the RBD fusion proteins retain antigenicity, induce a robust neutralizing antibody response and confer protection to immunized mice against SARS-CoV-2 challenge. While the present study focuses on SARS-CoV-2, the designer lambda system is exceedingly nimble and can be rapidly adapted for any established or emerging pathogenic target, highlighting its potential as an effective vaccine platform.

## 2. Materials and Methods

### 2.1. Materials

A pcDNA3.4 Topo vector modified to contain a human immunoglobulin heavy chain signal sequence and an N-terminal tag consisting of hexahistidine and SUMOstar sequences was a generous gift from Drs V. Upadhyay and K.M.G. Mallela, CU Anschutz Medical Campus [15]. We refer to this vector as ‘pMTopo’ and the N-terminal tag as ‘H6S-’ with the dash indicating that this portion can be proteolytically cleaved. They also kindly provided us with pMTopo-SCV2.RBD, an expression vector for wildtype, full-length SARS-CoV-2 RBD (strain Wuhan-Hu-1, S protein residues R319-F541) (GenBank, MN908947.3), and an expression vector for SUMOstar protease.

Expi293F^TM^ cells derived from the human embryonic kidney 293 (HEK293) cell line (Cat. #A14527), Expi293^TM^ expression medium (A1435101), OptiPRO^TM^ serum-free medium (12309019) and Nalgene™ Single-Use PETG Erlenmeyer Flasks (plain bottom, vented closure; 4115-0125) were purchased from Thermo Fisher Scientific (Waltham, MA, USA). Transfection-grade linear polyethylenimine hydrochloride (PEI MAX^®^, MW 40k; 24765-1) and Whatman^®^ GD/X syringe filters (WHA69002502) were purchased from Polysciences (Warrington, PA, USA) and Millipore Sigma (Burlington, MA, USA), respectively. Amicon centrifugal filters (3-100k MWCO) were purchased from Sigma-Aldrich (St. Louis, MO, USA).

All chromatography columns employed (HiTrap Q HP, HiTrap SP, HisTrap FF, Superose 6 Increase) were purchased from GE Healthcare (Marlborough, MA, USA). Protein purifications utilized an ÄKTA Purifier chromatography system (GE Healthcare), and absorbance spectra were obtained using a NanoDrop UV−Vis spectrophotometer (Thermo Fisher Scientific Inc.).

### 2.2. Protein Production

#### 2.2.1. Cloning, Expression and Purification of the RBD Constructs

Amino acid sequences of the lambda gpD protein (herein referred to as “D”) and SARS-CoV-2, strain Wuhan-Hu-1 S protein were obtained from UniProt (IDs: P03712, P0DTC2). DNA fragments were then designed to encode D and RBD (S protein residues N331-K528) connected by a flexible, 15 residue peptide linker [(GGGGS)_3_]. The nucleotide sequences were codon optimized for expression in mammalian cells and synthesized by Twist Biosciences (South San Francisco, CA, USA). DNA was independently cloned into pMTopo to afford the plasmids pMTopo-RBD::D and pMTopo-D::RBD, which were then used with PEI MAX^®^ and OptiPRO^TM^ to transfect Expi293F^TM^ cells. 

Transiently transfected cells were cultured under standard conditions (37 °C, 8% CO_2_, 135 rpm) in expression medium for 4–6 days, and secreted protein (H6S-RBD, H6S-RBD::D or H6S-D::RBD) was harvested and purified by immobilized metal ion affinity chromatography (IMAC). Briefly, the supernatant was filtered (0.22 μm), diluted 2-fold in 20 mM sodium phosphate [pH 7.4] buffer containing 500 mM NaCl and 20 mM imidazole and loaded onto a 5 mL HisTrap FF column equilibrated with the same buffer. Tagged protein was eluted with a linear gradient to 280 mM imidazole, and fractions containing the protein of interest were pooled and buffer exchanged into 50 mM sodium phosphate [pH 7.6 (H6S-RBD)/6.6 (H6S-RBD::D, H6S-D::RBD)] buffer containing 100 mM NaCl, 50 mM arginine, 50 mM glutamate, 0.2 mM TCEP and 1 mM EDTA for short-term (<1 week) storage at 4 °C. For long-term storage at −80 °C, stocks were supplemented with 20% glycerol. The purity of the preparations was determined to be >95% by sodium dodecyl-sulfate polyacrylamide gel electrophoresis (SDS-PAGE) (Appendix A), and the protein concentrations were calculated spectrally using an extinction coefficient at 280 nm of 36,580 M^−1^ cm^−1^ (H6S-RBD) or 52,050 M^−1^ cm^−1^ (H6S-RBD::D, H6S-D::RBD) as determined by the Edelhoch method with the Pace–Gray correction of the molar extinction coefficient equation for native proteins in water [16,17].

The purified tagged proteins were buffer exchanged into 50 mM Tris-HCl [pH 8.0] buffer containing 20 mM NaCl and digested with SUMOstar protease (60:1 molar ratio). The reaction mixture was supplemented with 5 mM DTT, incubated overnight at 4 °C and purified by IMAC, *vide supra*. Untagged proteins (RBD, RBD::D, D::RBD) recovered from the flowthrough were buffer exchanged into the 50 mM sodium phosphate [pH 7.6/6.6] buffer for short-term storage (4 °C) or supplemented with 20% glycerol for long-term storage (−80 °C). The protein concentration was determined spectrally (extinction coefficient [280 nm]: 33,600 M^−1^ cm^−1^, RBD; 49,070 M^−1^ cm^−1^, RBD fusions), and the purity of the preparations was determined to be >95% by SDS-PAGE.

#### 2.2.2. Purification of Phage Lambda Proteins

The lambda D protein was expressed in *E. coli* BL21(DE3)[pD] cells and purified by ammonium sulfate precipitation and ion exchange chromatography (IEC), as described [12,14] with modification. Following loading onto three HiTrap SP columns connected in tandem, bound protein was eluted with a linear gradient to 1 M NaCl. Fractions containing D were pooled and buffer exchanged into 75 mM HEPES [pH 7.4] buffer containing 140 mM NaCl for short-term storage (4 °C) or supplemented with 20% glycerol for long-term storage (−80 °C). The protein concentration was determined spectrally (extinction coefficient [280 nm]: 15,470 M^−1^ cm^−1^) and the purity was determined by SDS-PAGE (>98%) (Appendix A). 

Lambda PLPs were expressed in *E. coli* BL21(DE3)[pNu3_E] cells and purified by rate zonal centrifugation and IEC, as described [12,14]. The protein concentration was determined spectrally (extinction coefficient [280 nm]: 16.36 μM^−1^ cm^−1^) and the purity was determined by SDS-PAGE (>98%).

### 2.3. Biophysical Characterization Studies

#### 2.3.1. Analytical Ultracentrifugation

Sedimentation velocity analytical ultracentrifugation (SV-AUC) was performed using an Optima XL-A analytical centrifuge (Beckman Instruments Inc.; Fullerton, CA, USA) with an AN-60 Ti 4-hole rotor. Studies were conducted at 20 °C with a rotor speed of 50,000 rpm. Purified protein samples were buffer exchanged into 10 mM sodium phosphate [pH 6.6] buffer containing 1.8 mM KH_2_PO_4_, 137 mM NaCl and 2.7 mM KCl. Stock concentrations were determined spectrally, and samples were diluted to three protein concentrations spanning 10.5–63.3 μM (0.12–3.01 mg/mL) and loaded into 12 or 3 mm pathlength centerpieces (two-sector, charcoal-Epon) equipped with quartz windows; in all cases, the sodium phosphate buffer served as the reference. Absorbance data were collected at 250, 280 or 295 nm in a continuous scan mode (step size, 0.003 cm; scan density, 300 scans; time interval, ~206 s). The partial specific volume of each protein along with the solvent density and viscosity were calculated by SEDNTERP v.3.0.4, and the raw data were analyzed using SEDFIT v.16.1c and SEDPHAT v.15.2b [18].

#### 2.3.2. Circular Dichroism Spectroscopy

Circular dichroism (CD) spectra were recorded on a Chirascan Plus spectrometer (Applied Photophysics Inc.; Charlotte, NC, USA). Purified protein samples were buffer exchanged into 10 mM sodium phosphate [pH 6.7] buffer and analyzed at the indicated protein concentration in a 0.5 mm pathlength cuvette. Data at 20 °C were collected in 8 replicate scans at an interval of 1 nm (1.0 step) using a bandwidth of 1 nm and signal averaging time of 2 s. Secondary structure content was calculated from the CD spectroscopic data following deconvolution by single spectrum analysis with BeStSel [19] or using the CDSSTR algorithm (Set 3) with the DichroWeb server [20].

Thermally induced protein denaturation was monitored by far-UV CD spectroscopy. The thermal melts were recorded using a temperature ramp of 20–95 °C at a scan rate of 1 °C/min. Signals collected at 208, 222 and 230 nm were averaged for 2 s, and the data were converted to fraction unfolded (*F_U_*), plotted against the temperature (*T* [°C]) and fit to the following equation:(1)$$F_{U}=\frac{{(b}_{u}+m_{u}\cdot T)-(b_{l}+m_{l}\cdot T)}{1+{\left(\frac{T}{T_{M}}\right)}^{m_{T}}}+(b_{l}+m_{l}\cdot T)$$
where *b_u_* and *m_u_* are the y-intercept and slope of the upper baseline, respectively; *b_l_* and *m_l_* are the y-intercept and slope of the lower baseline; *m_T_* is the slope of the unfolding transition; and *T_M_* is the melting temperature.

### 2.4. In Vitro Phage Lambda PLP Studies

#### 2.4.1. Particle Decoration

Lambda PLPs were decorated *in vitro*, as described [14] with modification. Briefly, 40 nM PLPs were expanded in 10 mM HEPES [pH 7.4] buffer containing 2.5 M urea for 30 min on ice and then buffer exchanged into 10 mM HEPES [pH 7.4] buffer containing 200 mM urea. Expanded shells (30 nM) were decorated with either RBD::D or D::RBD at the desired surface density percentage for 45 min, followed by a 30 min incubation with D to fully saturate the shell. The reaction was performed in 10 mM HEPES [pH 7.4] buffer containing 20 mM arginine, 35 mM urea and 0.1% Tween 20 at room temperature. PLPs decorated at 10%, 25% or 50% surface density display 42, 105 or 210 copies of the RBD fusion, respectively, equivalent to approximately 1.26 μM (0.03 mg/mL), 3.15 μM (0.08 mg/mL) or 6.30 μM (0.16 mg/mL) RBD.

Decorated PLPs were purified by size exclusion chromatography (SEC), employing a 24 mL Superose 6 Increase 10/300 GL column and mobile phase of 25 mM HEPES [pH 7.6] buffer containing 100 mM NaCl, 50 mM arginine, 50 mM glutamate, 0.03% Tween 20 and 5% glycerol. Fractions containing decorated PLPs were pooled and buffer exchanged into 50 mM HEPES [pH 7.4] buffer containing 100 mM NaCl and 10 mM MgCl_2_ for storage at 4 °C. Purified proteins used in decoration reactions were also independently assessed by SEC under buffer conditions that matched those of the SV-AUC studies [10 mM sodium phosphate, 1.8 mM KH_2_PO_4_, 137 mM NaCl, 2.7 mM KCl_pH6.6].

#### 2.4.2. Particle Characterization

Physiochemical particle characterization based on light scattering was performed using a Zetasizer Nano ZS instrument (Malvern Panalytical Ltd., Malvern, UK) as described [12,14]. Electron micrographs were acquired on a 120 kV Thermo Fisher/FEI Talos L120C microscope with a Ceta CMOS detector. Sample preparation and particle analysis using Fiji [21] was performed as described [12,14].

### 2.5. Viruses and In Vivo Murine Studies

#### 2.5.1. Virus and Cell Culture

Vero E6 cells (ATCC, CRL-1586) were cultured at 37 °C in HyClone Dulbecco’s Modified Eagle Medium (HC-DMEM; Thermo Fisher Scientific, 11965-084) supplemented with 10% fetal bovine serum (FBS), 10 mM HEPES [pH 7.3], 1 mM sodium pyruvate, 1× non-essential amino acids and 100 U/mL of penicillin–streptomycin. SARS-CoV-2, isolate USA-WA1/2020 (BEI Resources, NR-52281) and strain MA10 were passaged once in Vero E6 cells, and the virus was then titrated by focus formation assay (FFA) and plaque assay, as described [22]. Work with infectious virus was performed at an Institutional Biosafety Committee approved BSL3 facility within the CU School of Medicine using positive pressure air respirators and other personal protective equipment.

#### 2.5.2. Animal Study Design

Animal studies were conducted in accordance with the Guide for the Care and Use of Laboratory Animals of the National Institutes of Health. Female BALB/c mice were purchased from The Jackson Laboratory (Bar Harbor, ME, USA), and mice aged 6–8 weeks were immunized with WT PLPs (control) or different versions of D::RBD PLPs diluted in 1× PBS [pH 7.4] via intramuscular (i.m.) injection (50 μL, hind leg). Mice were boosted 3 weeks after the primary immunization using the same dose and route. For virulent virus challenge, mice were anesthetized by intraperitoneal (i.p.) injection of a mix of xylazine (7.5 mg/kg) and ketamine hydrochloride (80 mg/kg) diluted in 1× PBS [pH 7.4]. Immunized mice were inoculated intranasally (i.n.) with 10^4^ PFU of SARS-CoV-2, strain MA10. Mice were monitored daily for signs of disease and weight recorded. At 4 days post-inoculation (dpi), animals were euthanized by exposure to isoflurane vapors followed by bilateral thoracotomy and tissues were collected for virological and immunological analyses.

#### 2.5.3. Particle Antigenicity and Immunogenicity Testing

The antigenicity of the PLP preparations was assessed by enzyme-linked immunosorbent assay (ELISA). Immulon 4HBX plates were coated with 0.2 μg of recombinant SARS-CoV-2 RBD protein (rRBD; strain Wuhan-Hu-1; GenPept, QHD43416), 0.2 μg of WT PLPs (control) or 0.2 μg of 10%, 25% or 50% RBD::D PLPs overnight at 4 °C. Plates were blocked with 3% nonfat milk in PBS-T (1× PBS, 0.1% Tween 20) for 1 h at room temperature. Wells were then washed (PBS-T, 3×) and incubated for 1.5 h at room temperature with chimeric human anti-SARS-CoV spike antibody (clone CR3022; Absolute Antibody NA; Shirley, MA, USA; Ab01680) prepared in a 2-fold dilution series in PBS-T (starting dilution, 1:200). The primary antibody was removed, and wells were washed (PBS-T, 3×) and probed with the secondary antibody diluted at 1:4000 in PBS-T. The detection antibody was removed, wells were washed (PBS-T, 3×) and plates were developed using 3,3′,5,5′-tetramethylbenzidine (Sigma-Aldrich, T0440, Massachusetts, United States). Following the addition of 0.3 M H_2_SO_4_, absorbance at 450 nm was measured using a Tecan Infinite M Plex plate reader (Tecan Group Ltd., Seestrasee, Switzerland).

SARS-CoV-2 RBD-specific antibody responses following immunization with WT PLPs, H6S-D::RBD PLPs or D::RBD PLPs were measured by ELISA using mouse sera. This was performed as previously described except that wells were probed with serum samples diluted in PBS-T supplemented with 1% nonfat milk for 1.5 h, following the blocking and washing steps. Samples were removed, and wells were washed (PBS-T, 3×) and probed with goat anti-mouse IgG-HRP (Southern Biotech; Birmingham, AL, USA; 1030-05) diluted at 1:4000 in PBS-T. Endpoint titers are reported as the reciprocal of the final dilution.

#### 2.5.4. Neutralization Assay

Neutralizing antibodies against SARS-CoV-2 were measured by a focus reduction neutralization test (FRNT), as described [14]. Briefly, Vero E6 cells were seeded in 96-well plates at 10^4^ cells/well. The next day heat-inactivated (56 °C, 30 min) serum samples were serially diluted (2-fold, starting at 1:10) in HC-DMEM supplemented with 1% FBS in 96-well plates. SARS-CoV-2 isolate USA-WA1/2020 (100 focus-forming units) was added to each well, and the plates were incubated (37 °C, 1 h). Culture medium was then removed and replaced with the serum–virus mixture. Following incubation (37 °C, 1 h), samples were removed and cells overlaid with 1% methylcellulose (Millipore Sigma, M0512, Massachusetts, United States) in Minimum Essential Medium (MEM; Gibco-Thermo Fisher Scientific, 12000-063,) supplemented with 10 mM HEPES, 2% FBS and 100 U/mL of penicillin–streptomycin for 24 h at 37 °C. After fixing cells with 1% paraformaldehyde (Acros Organics-Thermo Fisher Scientific, 416780030, Waltham, MA, USA), cells were probed with 1 μg/mL of chimeric human anti-SARS-CoV spike antibody in Perm Wash (1× PBS, 0.1% saponin, 0.1% BSA) for 2 h at room temperature. Cells were washed (PBS-T, 3×) and incubated with goat anti-human IgG Fc-HRP (Southern Biotech, 2014-05, Birmingham, AL, USA) diluted at 1:1000 in Perm Wash for 1.5 h at room temperature. Viral antigen positive foci were revealed with TrueBlue substrate (SeraCare; Milford, MA, USA; 5510-0030) and counted using a BioSpot analyzer (Cellular Technology Ltd., Cleveland, OH, USA). FRNT_50_ titers were calculated as previously described [14].

#### 2.5.5. Quantification of SARS-CoV-2 Genomic and Subgenomic RNA

To quantify viral genomic and subgenomic RNA, lung tissue of mice challenged with SARS-CoV-2, the strain MA10 was homogenized in TRIzol reagent (Life Technologies-Thermo Fisher Scientific, 15596018) and total RNA was isolated using a PureLink RNA Mini kit (Life Technologies-Thermo Fisher Scientific, 12183025). Single-stranded cDNA was generated using random primers and SuperScript IV reverse transcriptase (Life Technologies-Thermo Fisher Scientific, 18091050). Virus RNA copies (genomic/subgenomic) were measured by qPCR using the primer and probe combinations provided in [14]. Genomic RNA levels were quantified using a standard curve generated from known FFU of SARS-CoV-2, as described [23]. Subgenomic RNA levels were extrapolated from a standard curve using defined concentrations of a plasmid encoding the SARS-CoV-2 subgenomic N fragment (pCR-sgN TOPO), as described [14]. All qPCR reactions were prepared with Taqman Universal MasterMix II (Applied Biosystems-Thermo Fisher Scientific, 4440038) and examined using an Applied Biosystems QuantStudio ViiA 7 analyzer.

### 2.6. Graphing and Statistical Analyses

Plots were rendered using GraphPad Prism v.9/10.1.2 (GraphPad Software; Boston, MA, USA). Statistical significance was assigned when *p*-values were <0.05 using the same software, and the corresponding information (*e.g.*, statistical test, number of samples/test groups (n), median values, statistical comparison groups) is elaborated in the respective figure caption.

## 3. Results

### 3.1. Interrogation of SARS-CoV-2 and Phage Lambda for RBD Fusion Design

Figure 1 shows single particle models of the SARS-CoV-2 virion (Nanographics GmbH, v.2020.05.17; Vienna, Austria) and the phage lambda capsid with only the D network displayed [24]. The SARS-CoV-2 particle measures 100 ± 19 nm in diameter, has a spherical to ellipsoidal geometry and is coated with 20–40 heavily glycosylated S trimeric spikes that are randomly distributed on the virion and protrude ~25 nm from the surface [25,26,27,28]. Conversely, a lambda PLP decorated entirely with wildtype D (WT PLP) measures 70 ± 2 nm in diameter, has an icosahedral geometry and contains 140 D trimeric spikes that are uniformly distributed on the shell and protrude ~6 nm from the surface [12,24]. Fusion proteins used in phage display applications have been shown to project the ligand from the shell surface without interfering with D-binding or trimerization.

### 3.2. Engineering the RBD Constructs

The lambda decoration protein has been used in phage display applications with both N- and C-terminal fusions efficiently presenting the target ligand [29]. Therefore, two fusion proteins were engineered with SARS-CoV-2 RBD positioned at either the N-terminus (RBD::D) or C-terminus (D::RBD) joined by a 15 residue, flexible linker to allow for a high degree of movement to each protein [30] (Figure 2a). Only residues N331-K528 of RBD were included in the design of the RBD fusions, but this conserves the two canonical N-linked glycosylation sites and eight paired cysteine residues required for RBD structure stabilization [31].

Shown in Figure 2b are the structural models of the chimeric fusion proteins generated using AlphaFold 3 [32]. From this, the RBD fusions are noted to possess predominantly coil/loop secondary structures with protein cores that are stabilized by beta sheets. Consistent with this observation, secondary structure prediction analysis (sequence-based, SABLE server [33]) estimates that the RBD constructs (RBD, RBD fusions) will exhibit a 64% coil structure (7% helix, 30% beta strand/sheet) (Appendix A; Appendix A).

### 3.3. RBD Construct Characterization

The RBD construct preparations were determined to be 98% pure by SDS-PAGE (Figure 2c and Appendix A). Analytical characterization by SEC demonstrates that both RBD fusions elute as a single major species with an overall narrow elution profile but show a slight tendency to form high-molecular weight species in the concentration range of 21–63 μM (Figure 3a and Appendix A). Hydrodynamic parameters for the major species derived from the SEC analysis are summarized in Appendix A. As an orthogonal approach, SV-AUC was employed to quantitatively assess the proteins. Data collected using a 6-fold dilution series (11–63 μM) reveal *c*(*s*) size distributions that indicate the proteins are characterized by a single predominant species that is monodisperse (Figure 3b and Appendix A). The experimentally derived molar masses are consistent with the predicted molecular weight of a monomer based on amino acid composition (Appendix A).

Structural features of the fusion proteins were examined by CD spectroscopy, with RBD and D serving as reference controls. The far-UV CD spectra (185–260 nm) of the RBD fusions are characterized by maxima at ~193 nm and 231 nm and minima at ~208 nm (Figure 4a and Appendix A). These profiles closely match the theoretical spectrum with features that retain the random coil-character of D (positive band at 229 nm) and alpha-helical character of RBD (negative band at 208 nm, positive band at 193 nm) (Appendix A). Secondary structure content calculated from the CD spectroscopic data following deconvolution (Appendix A) closely matches the sequence-based prediction results (Appendix A). Thermal stability studies revealed that RBD fusions unfold in a cooperative manner and are of similar stability (Figure 4b and Appendix A). Unfortunately, unfolding was only partially reversible, precluding further thermodynamic analysis. Nevertheless, the unfolding data can still be analyzed to obtain a midpoint temperature (*T_M_*) that reflects the relative stability of the proteins (Appendix A).

### 3.4. RBD Fusions Are Efficiently Displayed on PLPs and Remain Antigenic

RBD fusion PLPs were generated by decorating shells with 10%, 25% or 50% of RBD::D or D::RBD as demonstrated in the analyses by agarose gel electrophoresis (AGE) and SDS-PAGE (Figure 5a,b). Electron micrographs of decorated, SEC-purified PLPs verified that the particles retain icosahedral symmetry and are characterized by surface projections that increase in density as the RBD fusion copy number is increased (Figure 5c). Physiochemical characterization of RBD fusion PLP preparations by dynamic and electrophoretic light scattering analyses (DLS, ELS) reveal a weighted mean hydrodynamic size (Z-Ave) of (98 ± 13, RBD::D) to (113 ± 5.7, D::RBD) d.nm and overall surface charge (ZP) of (−15 ± 1.4) to (−26 ± 1.4 mV) (Appendix A). The polydispersity index (PDI) for these formulations ranges from (0.23 ± 0.04) to (0.33 ± 0.11) (Appendix A). Lastly, RBD::D PLPs were assessed by enzyme immunoassays (ELISA) using a conformation-dependent, RBD-specific antibody to establish whether SARS-CoV-2 RBD retains antigenic properties when expressed as a fusion with D. Figure 5d demonstrates that the RBD::D PLPs are antigenic with a response that is nearly as robust as the control (recombinant RBD, rRBD).

### 3.5. H6S-D::RBD PLPs Are Immunogenic

We employed 10% H6S-D::RBD PLPs to evaluate the immunogenicity of the vaccine platform at a limiting, ‘low dose’ administration in vivo. For this, BALB/c mice were inoculated with 2.5 μg WT PLPs (control) or 10% H6S-D::RBD PLPs by i.m. injection and boosted by the same dose and route three weeks later. Serum was collected at 14 days post-primary immunization and 13 days post-boost (34 days post-prime), and RBD-specific IgG responses were quantified by ELISA (capture antigen, rRBD). Mice immunized with WT PLPs showed no reactivity to RBD in this assay, whereas a single immunization with 10% H6S-D::RBD PLPs resulted in a robust RBD-specific IgG response that was enhanced 2-fold following the booster immunization (Figure 6a). Similar results were observed in mice immunized with 25% H6S-D::RBD PLPs, 10% D::RBD PLPs and 25% D::RBD PLPs (Appendix A). The functionality of the RBD-specific antibody responses was evaluated by testing serum from immunized mice for the capacity to neutralize infectious SARS-CoV-2 by FRNT. Serum from mice immunized with WT PLPs showed no neutralization of the virus at either 14 or 34 days post-prime. Conversely, all mice that had received a single immunization with 10% H6S-D::RBD PLPs showed the capacity to neutralize live SARS-CoV-2 at 14 days post-prime, and neutralization was significantly enhanced (>10-fold) following the boost (Figure 6b). Again, similar results were observed in mice immunized with 25% H6S-D::RBD PLPs, 10% D::RBD PLPs and 25% D::RBD PLPs, although we note that mice immunized with 10% D::RBD PLPs had low neutralizing titers post-prime (Appendix A). These findings confirm that immunization with D::RBD PLPs (+/−, H6S-) decorated with as few as 42 copies of the fusion protein induces robust RBD-specific IgG antibody responses with neutralization capability and are consistent with our previous studies [14].

### 3.6. Vaccination with H6S-D::RBD PLPs Protects Against Virulent SARS-CoV-2 Challenge

Next, we evaluated the protective capacity of immunization with 10% H6S-D::RBD PLPs against a virulent SARS-CoV-2 infection using SARS-CoV-2, strain MA10. This mouse-adapted strain establishes a productive infection in the lungs of mice and results in disease manifestations consistent with COVID-19 in humans [22]. As anticipated, mice immunized with WT PLPs showed rapid weight loss following infection with peak loss occurring at 3 dpi—consistent with the viral infection model [22]. In contrast, mice immunized with 10% H6S-D::RBD PLPs showed no weight loss following infection (Figure 6c). Similar results were obtained for mice immunized with 25% H6S-D::RBD, although protection against weight loss after 2 dpi was more significant; in contrast, mice immunized with 10% or 25% D::RBD PLPs were not protected against weight loss (Appendix A). At 4 days post-intranasal inoculation, mice were euthanized, and lung tissue was collected for viral burden analysis by RT-qPCR. Immunization with 10% H6S-D::RBD PLPs resulted in a significant reduction in both genomic and subgenomic viral RNA of the lung tissue compared to those from mice immunized with WT PLPs (Figure 6d). As before, similar results were obtained for mice immunized with 25% H6S-D::RBD but with the extent of protection against lung viral burden approximately equivalent; while mice immunized with 10% or 25% D::RBD PLPs were also protected against SARS-CoV-2 infection, the extent of protection against lung viral burden was less robust (Appendix A).

## 4. Discussion

Several nanoparticle systems have been described wherein surface proteins are modified either chemically or genetically to display heterologous protein ligands on the particle surface. Ferritin [34], “spy-catcher” technologies [35,36,37] and a number of phage platforms [38,39,40,41,42] and others have been described. Several of these systems have been developed as vaccine platforms and while all have strengths and weaknesses, lambda PLPs provide an ensemble of demonstrated advantages and unique features that can be harnessed for vaccine development [10,14]: (i) lambda PLPs can be decorated with up to 420 ligands per shell under rigorously defined *in vitro* conditions - a significant increase in display density compared to other platforms; (ii) the PLPs can be decorated with multiple ligands, alone or in combination, including biologics (proteins, DNA, carbohydrates, antibodies), small molecules (drugs, fluorophores) and synthetic polymers (PEG, hyaluronic acid); (iii) engineering of the lambda platform is fast and tunable; (iv) the particle is self-adjuvanting [14,43]; and (v) decorated lambda PLPs have been shown to be monodisperse, stable and possess physiochemical properties amenable to pharmaceutical development and formulation. This demonstrated combination of features is unique and affords an innovative and adaptable platform for vaccine development. 

The “second generation” platform described herein employs genetic fusion display ligands, which streamlines the engineering process, compared to the chemical cross-linking approach described previously. Expression of the fusion constructs in eukaryotic cells ensures appropriate glycosylation of the RBD, which can then be used to efficiently decorate the PLP *in vitro*. The ensemble of biochemical, biophysical, structural and physiochemical data indicate that (i) RBD fusion proteins are folded and stable monomers in solution; (ii) PLPs decorated with either RBD::D or D::RBD afford particles that retain wildtype geometry but display a density protruding farther from the shell surface consistent with the increased size of the RBD fusions; (iii) the preparations have a high degree of homogeneity that is acceptable by pharmaceutical standards; and (iv) the RBD fusion PLPs retain antigenicity. While the PLPs can tolerate ≥50% occupation of the D-binding sites with the RBD fusion proteins, decoration with 25% of RBD::D (105 copies per particle) affords a maximal antigenic response. Furthermore, immunization with 10% H6S-D::RBD PLPs (displaying only 42 copies per particle) leads to the generation of a robust RBD-specific IgG response with neutralizing capacity and confers significant protection from virulent SARS-CoV-2 infection in mice. Importantly, the surface density of the RBD (*e.g.*, dose) can be readily tuned to allow simultaneous optimization of the cost of vaccine production, formulation of the preparation and the immunological response. This allows for decoration with additional antigens (up to 378 copies per particle for a total of 420) to engineer a multi-valent vaccine, as previously demonstrated with our first-generation bi-valent platform [14].

## 5. Conclusions

These findings examine the evolution of the lambda PLP platform for the purpose of vaccine development. Specifically, we engineered chimeric proteins by genetically modifying the lambda decoration protein to contain SARS-CoV-2 RBD at either the N- or C-terminus to display the antigen on phage-like particles. Physiochemical characterization of the particles confirmed that they are stable, soluble and monodisperse—properties that are required for pharmaceutical development. The RBD fusion PLPs induce an immune response in mice that results in neutralizing antibody production and confers protection against virulent virus challenge. The results confirm the therapeutic utility of the second-generation platform and provide important information regarding the development of countermeasures against circulating and newly emerging pathogens.

## Figures and Tables

**Figure 1 vaccines-12-01201-f001:**
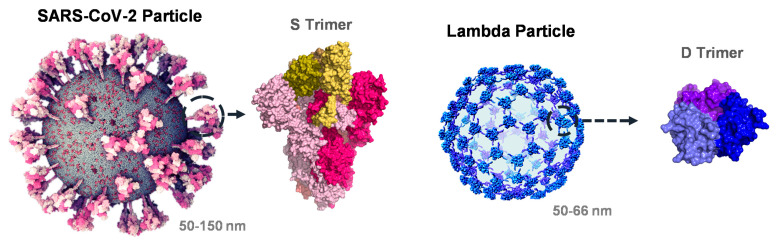
**Molecular architecture of the viral particles.** Structural comparison of the SARS-CoV-2 virion (Nanographics GmbH, v.2020.05.17) and bacteriophage lambda capsid shell with only the D network shown for clarity [24]. Homotrimeric spikes of the S (PDB 6VSB) and D (PDB 1C5E) proteins are shown from the side, colored in shades of pink and blue, respectively. The RBDs (R319-F541) in the S trimer have been highlighted in shades of yellow.

**Figure 2 vaccines-12-01201-f002:**
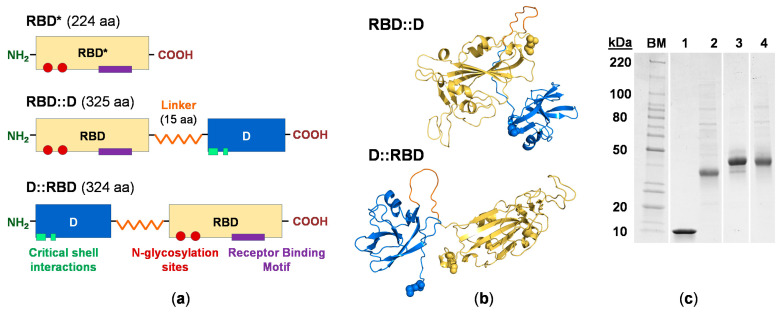
**Protein design and purification.** (**a**) Schematic of the RBD constructs. The RBD protein construct (RBD*) is the wildtype, full-length receptor-binding domain in the S1 fragment of the SARS-CoV-2 spike glycoprotein; in the RBD fusions (RBD::D, D::RBD), this domain has been slightly truncated. RBD and D proteins are connected by a large, flexible linker, respectively colored as yellow, blue and orange. For RBD, the N-glycosylation sites and residues spanning the receptor binding motif are indicated as red circles and purple rectangle, respectively; whereas residues of the D protein involved in lambda shell interactions are shown in turquoise. See Appendix A (Appendix A for sequence details, and Appendix A for the schematic including the cleavable tag (H6S-)). (**b**) Structural models of the RBD fusions were generated using AlphaFold3 colored as in (**a**). The N- and C-termini in both models are rendered as spheres. (**c**) Purified proteins analyzed by SDS-PAGE, stained with Coomassie Blue. Lanes: BM, BenchMark Protein Ladder (10–220 kDa); 1, D (11.6 kDa); 2, RBD (23.4 kDa); 3, RBD::D (34.9 kDa); 4, D::RBD (34.8 kDa). Molecular weight estimates are based on amino acid composition. See Appendix A for a complete gel with tagged proteins included. Abbreviations: SARS-CoV-2, severe acute respiratory syndrome coronavirus 2; RBD, receptor-binding domain; D, lambda decoration protein.

**Figure 3 vaccines-12-01201-f003:**
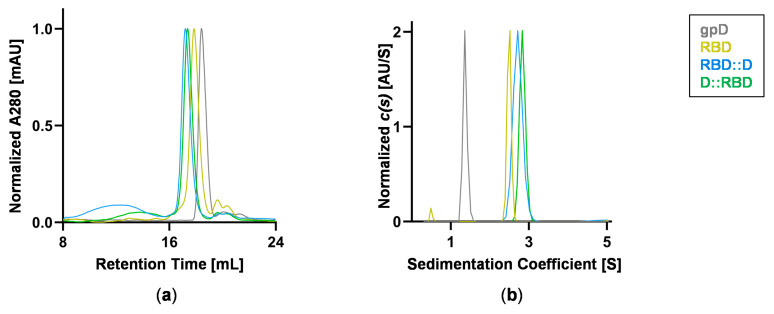
**Protein biophysical characterization.** Purified proteins were analyzed by size exclusion chromatography (SEC) and sedimentation velocity analytical ultracentrifugation (SV-AUC). Proteins (21 μM, 0.2–0.7 mg/mL): grey, D; yellow, RBD; blue, RBD::D; green, D::RBD. (**a**) Overlay of SEC chromatograms with traces representing an average of three independent injections with the signal (A280) normalized to 1. Similar results were acquired at 63 μM, and hydrodynamic parameters are provided in Appendix A. (**b**) Overlay of SV-AUC sedimentation coefficient distributions. The best-fit *c(s)* distributions were normalized to 1. *c*(*s*) analysis of the full concentration series is provided in Appendix A with fit parameters presented in Appendix A.

**Figure 4 vaccines-12-01201-f004:**
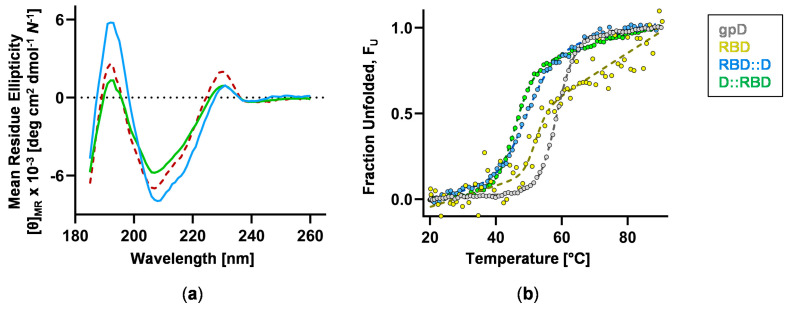
**Assessment of protein secondary structure and thermal stability.** Purified proteins were analyzed by circular dichroism (CD) spectroscopy. Proteins: grey, D (35 μM, 0.4 mg/mL); yellow, RBD (4 μM, 0.1 mg/mL); blue, RBD::D (6 μM, 0.2 mg/mL); green, D::RBD (12 μM, 0.4 mg/mL). (**a**) Overlay of far-UV CD spectra acquired at 20 °C and plotted in units independent of polymer length ([θ]_MR_, mean residue ellipticity). A theoretical spectrum (red, dashed line) was generated by averaging the experimentally acquired spectra of D and RBD. Refer to Appendix A for additional data. (**b**) Thermal denaturation profiles, continuously monitored by far-UV CD at 230 nm over the range of 20–95 °C and plotted as fraction unfolded against temperature. The data were fit to Equation 1 to obtain the *T_M_* for each protein (see Appendix A).

**Figure 5 vaccines-12-01201-f005:**
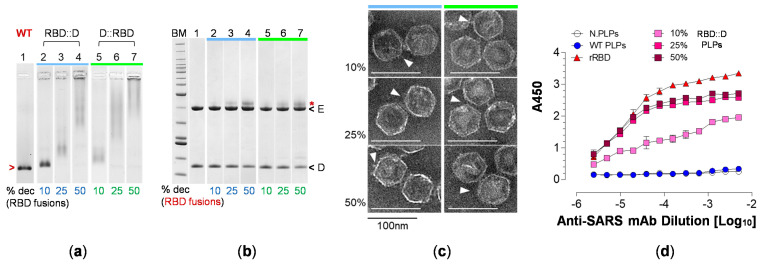
**PLP decoration with fusion constructs and confirmation of antigenicity.** PLPs were decorated with RBD::D or D::RBD as indicated by the blue and green bars. (**a**) Analysis of the reaction mixtures by AGE following Coomassie Blue staining. Lanes: 1 (WT), WT PLPs; 2–4 (RBD::D), PLPs displaying 10%, 25% and 50% RBD::D; 5–7 (D::RBD); PLPs displaying 10%, 25% and 50% D::RBD. The migration of the reference standard (WT PLPs) is indicated in red. (**b**) Analysis of the reaction mixtures by SDS-PAGE following Coomassie Blue staining. Lanes: BM, BenchMark Ladder (10–220 kDa); 1, WT PLPs; 2–4, 10%, 25% and 50% RBD::D PLPs; 5–7, 10%, 25% and 50% D::RBD PLPs. The migration of the lambda major capsid and decoration proteins (gpE, gpD) is indicated as ‘E’ and ‘D’, respectively. The red asterisk denotes the migration of the RBD fusions. (**c**) Electron micrographs of RBD fusion PLPs with the surface density attributed to the chimeric protein indicated by white arrows (magnification, 45,000×; scale bars, 100 nm). (**d**) Particle immunoreactivity by ELISA using 0.2 μg recombinant RBD (rRBD) or 0.2 µg RBD::D PLPs at three different surface density percentages (10%, 25% and 50% RBD::D - approximately equivalent to 0.013, 0.031 and 0.057 μg of free RBD, respectively). Naked PLPs and WT PLPs serve as experimental controls.

**Figure 6 vaccines-12-01201-f006:**
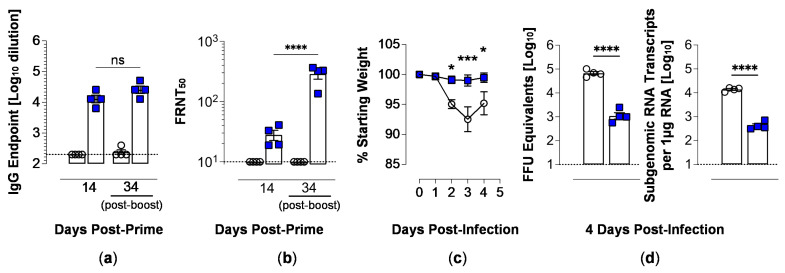
**Immunization with H6S-D::RBD PLPs results in protection from SARS-CoV-2 infection.** BALB/c mice were immunized with 2.5 μg of WT PLPs (control; open circles) or 10% H6S-D::RBD PLPs (blue squares) via intramuscular inoculation (*n* = 4 mice/group). At 21 days post-prime, mice received a boost at the same dose and route. (**a**,**b**) Serum samples collected at 14 and 34 days post-primary (13 days post-boost) immunization were analyzed for total anti-RBD IgG by ELISA (**a**) and neutralizing activity against live SARS-CoV-2 by a focus-reduction neutralization test (FRNT) (**b**). Error bars represent the mean ± SEM. *p*-values determined by one-way ANOVA with Tukey’s multiple comparisons test: ns (not significant), **** *p* < 0.0001. (**c**) At 35 days post-prime (14 days post-boost), mice were challenged intranasally with 10^4^ PFU of SARS-CoV-2, strain MA10 and weight was monitored daily. Error bars represent the mean ± SEM. *p*-values were determined by two-way ANOVA with Sidak’s multiple comparisons test: * *p* < 0.05, *** *p* < 0.001, **** *p* < 0.0001. (**d**) At 4 dpi, lung tissue viral burden was quantified by RT-qPCR for viral genomic RNA (*left*) and N subgenomic RNA (*right*). *p*-values were determined by unpaired students *t*-test: **** *p* < 0.0001. Immunizations using PLPs decorated with 25% H6S-D::RBD, 10% D::RBD or 25% D::RBD yielded similar results (see Appendix A).

## Data Availability

The data supporting the findings of this study are available on request from the corresponding author.

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
