# Peer review of "Second-Generation Phage Lambda Platform Employing SARS-CoV-2 Fusion Proteins as a Vaccine Candidate"

_vaccines, 2024, doi:10.3390/vaccines12111201_

Round 1
Reviewer 1 Report
Comments and Suggestions for Authors
This manuscript presents the results of generation and in vitro and in vivo evaluation of a prototype vaccine against SARS-CoV-2 based on the lambda phage-like protein platform. The authors engineered chimeric proteins by fusing the lambda decoration protein with SARS-CoV-2 RBD at either the N- or C-terminus to display the antigen on phage-like particles. The experimental part is well-designed and well-performed, and the vaccine candidates were proved to be pure, stable and antigenically identical to the RBD protein. As expected, such vaccines have a potential to induce virus-neutralizing immune responses and protect mice against lethal SARS-CoV-2 challenge.
Major point:
The manuscript is well-written except the Discussion section. The authors should thoroughly discuss their proposed vaccine platform in the context of existing vaccine platforms, including its advantages and limitations.
Minor comments:
1. Figure 5a. Please add molecular weight ladder, and also specific % of decorated proteins to each lane
2. Why did you select only one construct (10% D::RBD PLPs) for mouse studies? Wouldn’t 25% decoration even more immunogenic and protective?
3. Was the vaccine well tolerated in mice? E.g., no weight loss post vaccination?
4. Did you measure antibody to the phage proteins? Will it interfere with subsequent vaccinations?
Author Response
Reviewer #1: Comments and Suggestions for Authors
This manuscript presents the results of generation and in vitro and in vivo evaluation of a prototype vaccine against SARS-CoV-2 based on the lambda phage-like protein platform. The authors engineered chimeric proteins by fusing the lambda decoration protein with SARS-CoV-2 RBD at either the N- or C-terminus to display the antigen on phage-like particles. The experimental part is well-designed and well-performed, and the vaccine candidates were proved to be pure, stable and antigenically identical to the RBD protein. As expected, such vaccines have a potential to induce virus-neutralizing immune responses and protect mice against lethal SARS-CoV-2 challenge.
Major point:
The manuscript is well-written except the Discussion section. The authors should thoroughly discuss their proposed vaccine platform in the context of existing vaccine platforms, including its advantages and limitations.
We have modified the Discussion section to include an expanded discussion of the lambda PLP platform in relation to existing platforms, as suggested (Pg. 17).
Minor comments:
- Figure 5a. Please add molecular weight ladder, and also specific % of decorated proteins to each lane.
The image in Figure 5a is that of a native agarose gel analysis of lambda PLPs, and a molecular weight ladder is not appropriate. In its stead, the “WT” lane is provided as a standard, consisting of the PLPs decorated with 100% D protein (indicated with an arrow). The specific % decoration has been added to both Figures 5a and 5b, as requested.
- Why did you select only one construct (10% D::RBD PLPs) for mouse studies? Wouldn’t 25% decoration even more immunogenic and protective?
We initially chose to directly examine the immunological properties of the “lowest dose” preparation to determine the efficacy of the platform. Moreover, future goals of this project include the development of mosaic PLPs that simultaneously present numerous antigens – an additional reason we chose to test the 10% decoration with a single antigen as this affords the opportunity to decorate PLPs with up to 10 distinct antigens. Indeed, the data demonstrate that display of only 42 copies of the RBD fusion protein per particle induces a robust neutralizing antibody response and confers protection to immunized mice against SARS-CoV-2 challenge.
Nevertheless, additional data has been added to the revised manuscript, including results using tagged fusion proteins (10% and 25% H6S-D::RBD PLPs) and untagged fusion proteins (10% and 25% D::RBD PLPs). The new data is described in Figure 6, in the text (pgs. 15, 16, 17, in red) and in new Supplemental Figure S4. This new data further emphasizes that the D::RBD-PLPs provide a nimble platform for vaccine development, the primary goal of the present manuscript.
- Was the vaccine well tolerated in mice? E.g., no weight loss post vaccination?
We note that phage platforms are noninfectious in humans and considered non-toxic to eukaryotic cells. Consistently, no evidence of toxicity of λ phage-based particles has been observed in our cell culture or animal studies. Indeed, mice immunized with up to 10 ug of PLPs displayed no adverse events for months after multiple PLP inoculations 1.
- Did you measure antibody to the phage proteins? Will it interfere with subsequent vaccinations?
We appreciate this reviewer concern. We have not directly measured antibody responses to phage proteins. However, as shown in this study, and our prior study 1, the RBD specific antibody response is increased following a second immunization with the decorated PLPs. These data indicate that an anti-phage protein antibody response does not inhibit the effects of a prime-boost immunization. However, we acknowledge that it remains to be determined if an anti-phage antibody response limits the immunogenicity of additional PLP vaccinations.
Literature Cited
- Davenport, B.J., et al. Phage-like particle vaccines are highly immunogenic and protect against pathogenic coronavirus infection and disease. npj Vaccines 7, 57 (2022).
- Leist, S.R., et al. A Mouse-Adapted SARS-CoV-2 Induces Acute Lung Injury and Mortality in Standard Laboratory Mice. Cell 183, 1070-1085 (2020).
- Catalano, C.E. Bacteriophage Lambda as a Nano Theranostic Platform. in Physical Virology: From the State-of-the-Art to the Future of Applied Virology (eds. Comas-Garcia, M. & Rosales-Mendoza, S.) 307-328 (Springer International Publishing, New York, 2023).

Reviewer 2 Report
Comments and Suggestions for Authors
The paper discusses the development of a second-generation vaccine platform using chimeric proteins from SARS-CoV-2 and bacteriophage lambda. This platform allows for the efficient display of multiple viral antigens on phage-like particles (PLPs), improving vaccine manufacturing potential. Immunization studies in mice showed that particles with as few as 42 copies of the chimeric protein elicited strong neutralizing antibody responses and protected against virulent SARS-CoV-2. The platform's versatility suggests that it can be adapted for other infectious diseases, underscoring its potential for rapid vaccine development.
Comments:
1. The copy number of RBD::D or D::RBD is 42-210. Why is there so much variation? This is not good for vaccine production. Is it possible to make it more uniform?
2. The authors expressed two fusion proteins, RBD::D and D::RBD, but only D::RBD PLPs were used in animal experiments. Should the authors explain why?
3. Regarding the efficacy of the D::RBD PLP vaccine, it would be great to provide the survival rate data in the challenge experiment. Otherwise, please explain why this was not done.
4. Figure 2b did not provide efficacy information. I would suggest removing it and moving Fig 2a to Fig 1. What is the meaning of the star in Fig. 5b?
Author Response
Reviewer #2: Comments and Suggestions for Authors
The paper discusses the development of a second-generation vaccine platform using chimeric proteins from SARS-CoV-2 and bacteriophage lambda. This platform allows for the efficient display of multiple viral antigens on phage-like particles (PLPs), improving vaccine manufacturing potential. Immunization studies in mice showed that particles with as few as 42 copies of the chimeric protein elicited strong neutralizing antibody responses and protected against virulent SARS-CoV-2. The platform's versatility suggests that it can be adapted for other infectious diseases, underscoring its potential for rapid vaccine development.
- The copy number of RBD::D or D::RBD is 42-210. Why is there so much variation? This is not good for vaccine production. Is it possible to make it more uniform?
The PLP platform can be decorated with up to 420 copies of a display ligand, which we define as 100% surface density. Our protocol allows for decoration at lower surface densities in a defined manner, such that the number RBD copies can be controlled. In the present study, we decorate the PLPs as follows:
|
RBD Surface Density |
Number of RBDs per PLP |
|
10% |
42 |
|
25% |
105 |
|
50% |
210 |
Thus, in each case the number of RBD copies is defined and uniform, not a range. Pardon the confusion, and this has been clarified in the text, accordingly (Figure 5 legend, Pgs. 7 and 14 in red text).
- The authors expressed two fusion proteins, RBD::D and D::RBD, but only D::RBD PLPs were used in animal experiments. Should the authors explain why?
The gpD decoration protein has been used in phage display applications for decades and both N-terminal and C-terminal display fusions have been employed. A priori, we did not know which RBD fusion construct would possess the most favorable physiochemical and immunological properties, and we therefore engineered both for the initial screen. The data clearly indicate that both constructs possess essentially identical physiochemical properties, they both decorate PLPs to afford particles with essentially identical structural/physiochemical properties and they both possess similar antigenic responses by ELISA. Thus, as explained above in response to R1, due to the scale of this study, we chose to directly examine the immunological properties of select constructs in vivo to determine the efficacy of the platform.
- Regarding the efficacy of the D::RBD PLP vaccine, it would be great to provide the survival rate data in the challenge experiment. Otherwise, please explain why this was not done.
Thank you for this comment. We note that SARS-CoV-2 MA10 infection results in limited mortality in adult immunocompetent BALB/c mice 2; substantial mortality in this model is only observed in aged immunocompetent BALB/c mice (i.e., 1-year old). Thus, in these studies, mice were humanely euthanized at 4 days post virus inoculation to enable an assessment of vaccine efficacy against virus-induced weight loss and viral replication in the lung.
- Figure 2b did not provide efficacy information. I would suggest removing it and moving Fig 2a to Fig 1. What is the meaning of the star in Fig. 5b?
While we have previously demonstrated that PLP decoration with a variety of ligands (proteins, synthetic polymers, small molecules) can be controlled 3, we did not rigorously quantify the mg % surface density of the fusion proteins in this study. That said, electron microscopy shows increasing surface density as anticipated (Fig. 5c) and the ELISA data demonstrate a dose-dependent response which is consistent with increased surface density (Fig. 5d).
The asterisk in Figure 5b denotes the migration of the D::RBD and RBD::D fusion constructs. The figure and caption have been modified for clarity.
Literature Cited
- Davenport, B.J., et al. Phage-like particle vaccines are highly immunogenic and protect against pathogenic coronavirus infection and disease. npj Vaccines 7, 57 (2022).
- Leist, S.R., et al. A Mouse-Adapted SARS-CoV-2 Induces Acute Lung Injury and Mortality in Standard Laboratory Mice. Cell 183, 1070-1085 (2020).
- Catalano, C.E. Bacteriophage Lambda as a Nano Theranostic Platform. in Physical Virology: From the State-of-the-Art to the Future of Applied Virology (eds. Comas-Garcia, M. & Rosales-Mendoza, S.) 307-328 (Springer International Publishing, New York, 2023).

Round 2
Reviewer 1 Report
Comments and Suggestions for Authors
The authors revised their manuscript according to the reviewers' comments and added the Discussion section to refer to other studies with similar platforms. I have no further comments.